# Optimizing the Selection of Mass Vaccination Sites: Access and Equity Consideration

**DOI:** 10.3390/ijerph21040491

**Published:** 2024-04-17

**Authors:** Basim Aljohani, Randolph Hall

**Affiliations:** 1Department of Industrial and Systems Engineering, University of Florida, Gainesville, FL 32603, USA; 2Epstein Department of Industrial and Systems Engineering, University of Southern California, Los Angeles, CA 90089, USA; rwhall@usc.edu

**Keywords:** facility location, accessibility, route selection, vaccine distribution, disease management

## Abstract

In the early phases of the COVID-19 pandemic, vaccine accessibility was limited, impacting large metropolitan areas such as Los Angeles County, which has over 10 million residents but only nine initial vaccination sites, which resulted in people experiencing long travel times to get vaccinated. We developed a mixed-integer linear model to optimize site selection, considering equitable access for vulnerable populations. Analyzing 277 zip codes between December 2020 and May 2021, our model incorporated factors such as car ownership, ethnic group disease vulnerability, and the Healthy Places Index, alongside travel times by car and public transit. Our optimized model significantly outperformed actual site allocations for all ethnic groups. We observed that White populations faced longer travel times, likely due to their residences being in more remote, less densely populated areas. Conversely, areas with higher Latino and Black populations, often closer to the city center, benefited from shorter travel times in our model. However, those without cars experienced greater disadvantages. While having many vaccination sites might improve access for those dependent on public transit, that advantage is diminished if people must search among many sites to find a location with available vaccines.

## 1. Introduction

Coronavirus disease 2019 (COVID-19) emerged as a global threat in late 2019, quickly evolving into a pandemic that deeply impacted modern life. The virus’s rapid spread across the globe, due to its transmissibility and initial absence of population immunity, placed massive pressure on healthcare systems, disrupted economies, and altered social norms.

In response to the crisis, researchers and pharmaceutical companies worldwide have developed vaccines and therapies to combat the virus and prevent severe illness, hospitalization, and death. As vaccines became available in early 2021, pharmaceutical companies, government agencies, and healthcare organizations needed to rapidly develop the capacity to manufacture, distribute, and safely administer vaccines to eligible people, free of charge and voluntarily. The initial shortage of capacity meant that vaccine eligibility, allocation, and access were all limited and prioritized. 

This paper examines one aspect of prioritization: the selection of mass vaccine administration sites (i.e., locations where people are vaccinated) within metropolitan regions. For example, in Los Angeles (LA) County, just nine sites initially served more than 10 million people [1]. While some people could travel to sites with relative ease, others resided more than an hour from their nearest site.

### 1.1. Research Objectives 

Our aim is to develop and assess a methodology to optimize the location of mass vaccine administration centers, taking into consideration impacts on different population groups. We consider household automobile availability, travel time, distances, costs, and disparities among ethnic groups, showing the distributional effects of solutions based on different objective functions. In addition, socioeconomic factors, environment, and healthcare infrastructure contribute to an uneven distribution of health risks and resources in regions. Areas with lower socioeconomic status experience more chronic disease and have more limited access to healthcare facilities [2]. To represent variations in the risk of severe disease, we incorporate the Healthy Places Index (HPI) [3] in our analysis, which serves as a metric of healthcare vulnerability in localities. Last, we consider the possibility that people will travel further than the nearest administration site when necessary to obtain a vaccine appointment (as occurred when COVID-19 vaccines first became available).

In sum, our aim is to develop methods and insights to inform public health policies and decisions that improve access to vaccines through mass administration sites, considering variation in access among different population groups. We utilize COVID-19 in LA County as a case study, but our goal is to develop insights that inform future interventions against highly transmissible diseases.

### 1.2. Literature Review

Optimization models have been utilized in the healthcare sector to improve patients’ accessibility as demonstrated in [4], optimize resource allocation, and enhance scheduling [5]. These applications are critical for the strategic planning and decision-making phase, especially during global pandemics. In the case of vaccine distribution and optimal selection of vaccine centers, optimization models can help determine the most effective ways to distribute the vaccines to achieve the maximum coverage and accessibility and reduce the impact of the pandemic. From our review, the most recent research on this topic has focused on COVID-19. 

#### 1.2.1. Vaccine Allocation

Spatial analysis and accessibility have been a major part of the research on vaccine allocation as it helps identify disparities in vaccination rates and areas with highly vulnerable population groups. A study by Mollalo and Tatar (2021) [6] employed a GIS-based approach to delve into the spatial heterogeneity of vaccination rates across US counties, using the Social Vulnerability Index (SVI) as a key metric. Their findings underscored the relationship between socioeconomic factors and vaccination rates, which could help achieve more targeted vaccine allocation plans. Moreover, Alemdar, Kaya, Çodur, Campisi, and Tesoriere (2021) [7] emphasized vaccine logistics and the selection of vaccine administration sites. They introduced a three-step method for site selection, which involved defining eight evaluation criteria recommended by advisory boards. These criteria were then weighted, leading to the designation of potential sites and creating a suitability map for service areas. Cheng, Tao, Lian, and Huang (2021) [8] evaluated spatial accessibility to urban medical facilities in China using a detailed transportation network. Utilizing Kriging interpolation and cluster analysis, they found most areas had poor accessibility, though regions near subways were found to be better. They suggested enhancing the transportation network for improved medical facility access.

#### 1.2.2. Optimization of COVID-19 Vaccine Center Locations 

COVID-19 vaccine center locations have emerged as a significant area of study since the global pandemic emerged in 2020. Studies have employed mathematical modeling techniques, incorporating various factors, such as travel times, distances, and operational costs to identify optimal locations that would help reduce the impact of the pandemic. A study by Bravo, Hu, and Long (2022) [9] emphasized reducing travel distances to improve vaccination rates, suggesting using retail pharmacies and Dollar stores as sites. This approach would, by their estimates, bring 25% more people within a kilometer of a vaccine location, potentially increasing vaccinations by 5%. Risanger et al. (2021) [10] introduced a unique function to calculate the fraction of the population willing to travel to vaccination sites. Their insights into travel behaviors for different trip distances could be instrumental in public health facility planning. In another study by Bertsimas et al. (2021) [11], the DELPHI epidemiological model was developed, integrating it with an optimization strategy for vaccine allocation. Using a compartmental disease model, they forecasted pandemic dynamics and assessed vaccination impacts based on vaccine efficacy. The authors estimated that their approach would enhance vaccination campaign effectiveness by 20%, potentially saving 4000 US lives in three months. 

Alghanmi et al. [12] surveyed various optimization models for selecting administration centers, focusing on minimizing travel times, distances, and related costs. Furthermore, Lusiantoro et al. (2022) [13] developed a bi-objective linear model, focusing on maximizing vaccine coverage and minimizing travel distance. When applied to Yogyakarta, Indonesia, the authors found that solely prioritizing high COVID-19 case areas led to suboptimal coverage, indicating a need to consider additional factors. Another multi-objective optimization model was proposed by Tang, Li, Bai, Liu, and Coelho (2022) [14], where the authors tried to optimize the operational costs of vaccine sites and the total travel distance for multi-period COVID-19 vaccination planning. A framework was also proposed to help decision-makers choose sites based on real-life limitations or preferences while optimizing the service level. The authors predicted a 9.3% decrease in operational costs and a 36.6% decrease in the total travel distance. 

At the municipality level, Cabanilla, Enriquez, Mendoza, and Mendoza (2022) [15] presented optimal locations of vaccine sites, where they considered existing public facilities, such as hospitals and schools, as potential sites. They divided the town into several smaller areas and assigned weights to densely populated and highly contagious areas with higher case counts. The weighting factors were then incorporated into a distance minimization objective function. A location-allocation model was developed by Faisal, Alshammari, Alotaibi, Alghanmi, Bamsagm, and Bin Yamin (2022) [16] to improve the allocation of COVID-19 vaccine centers in Jeddah, Saudi Arabia. The authors introduced a maximal coverage model with and without facility capacity constraints. They applied the model with different impedance cutoffs, which are the maximum travel times required from demand points to vaccine centers. Moreover, the authors explored the minimum number of facilities needed to satisfy all the demand points within the city by minimizing the overall transportation time and distance.

## 2. Materials and Methods

In this section, we formulated a mixed-integer programming (MIP) model that aimed to optimize locations of vaccine centers against a cost-minimization objective in a large metropolitan area. The model assumed that each zone in the studied region represented both a potential site for COVID-19 vaccine administration and a population group that needed to be served by at least one vaccination administration center. Our model was intended to represent a time horizon when vaccination sites remained static. We assumed that all vaccines available within any time period are administered within the time period.

### 2.1. Optimization Model

Based on the stated assumptions, our decision variables represent the binary decision for whether or not each zone contains a vaccination center and the total number of vaccine doses allocated to each center and, in turn, to the population in each zone. Our objective function represents the sum of three costs (1) weighted transportation cost from home to site and back, (2) weighted travel time from home to site and back, and (3) fixed costs of opening sites. The weights are scenario dependent, each representing a different prioritization scheme: (1) population only, (2) HPI, and (3) a COVID-19 vulnerability index. Constraints represent the total number of vaccine doses available and a strategy to allocate them per the selected prioritization method, a maximum and minimum number of vaccine centers, a budget constraint to ensure the cost of opening the sites does not exceed the allocated budget, and a constraint to link vaccine allocations to the actual assignment of areas to centers. Our model also utilizes a flexibility parameter, F, which specifies the number of centers individuals can choose (with equal likelihood) from for the administration of vaccines, as a way to capture the effects of site-specific limitations on the availability of vaccine appointments. 

#### 2.1.1. Model Sets, Parameters, and Decision Variables

Sets:

*T*: Set of time periods, *t*.

*A*: Set of all zip code areas, *i*.

*S*: Set of all potential sites, *j*.

Parameters:

*TB_ij_*: Travel time by transit between area *i* and site *j*.

*TC_ij_*: Travel time by car between area *i* and site *j*.

*DD_ij_*: Distance between area *i* and site *j*.

*O_i_*: Percentage of car ownership in area *i*.

*W_i_*: Priority assigned for area *i*.

*P_i_*: Population of area *i*.

*K*: Available budget for opening sites.

*Q_t_*: Available quantity of the vaccine at time *t*.

*TP*: Total population of LA county.

*C*: the cost of opening a site.

*F*: A flexibility parameter indicating the maximum number of sites people can select from.

*M*: A large positive number. 

*MS*: The maximum number of allowable sites. 

*LS*: The minimum number of allowable sites.

*VT*: The cost value of time.

*VD*: The cost of traveled distance.

*VB*: The cost of a public transit ticket.

Decision Variables: 

*D_j_*: 1 if site *j* is selected as a site, 0 otherwise.

*X_ij_*: 1 if area *i* is assigned to site *j*, 0 otherwise.

*V_ijt_*: Allocated vaccines from site *j* to area *i* at time *t*.

#### 2.1.2. Objective Function and Constraints

We sought to optimize the following formulation, as explained below.
(1)$$\begin{array}{cc} Min\;\overset{S}{\underset{j}{\sum }}D_{j}C+\overset{A}{\underset{i}{\sum }}\overset{S}{\underset{j}{\sum }} & \left(\frac{1}{F}X_{ij}P_{i}(O_{i}TC_{ij}+\left(1-O_{i}\right)TB_{ij})\times 2\times VT)+\overset{A}{\underset{i}{\sum }}\overset{S}{\underset{j}{\sum }}(\frac{1}{F}X_{ij}P_{i}DD_{ij}O_{i}\times 2\times VD\right)\\ & +\overset{A}{\underset{i}{\sum }}\overset{S}{\underset{j}{\sum }}(\frac{1}{F}X_{ij}P_{i}(1-O_{i})\times 2\times VB) \end{array}$$
which is subject to the following:(2)$$\overset{S}{\underset{j}{\sum }}V_{ijt}=\frac{P_{i}}{TP}\times Q_{t},\;\mathrm{for}\;i\in A,\;t\;\in T$$
(3)$$\overset{S}{\underset{j}{\sum }}D_{j}\leq MS$$
(4)$$\overset{S}{\underset{j}{\sum }}D_{j}\geq LS$$
(5)$$\overset{S}{\underset{j}{\sum }}X_{ij}\geq F,\;\mathrm{for}\;i\in A$$
(6)$$X_{ij}\leq D_{j},\;\mathrm{for}\;i\in A,\;j\in S$$
(7)$$\overset{S}{\underset{j}{\sum }}D_{j}C\leq K$$
(8)$$V_{ijt}\leq X_{ij}M,\;\mathrm{for}\;i\in A,\;j\in S,\;t\in T$$
(9)$$D_{j}\geq 0,\;\mathrm{for}\;j\in S$$
(10)$$X_{ij}=\left\{0,\;1\right\},\;\mathrm{for}\;i\in A,\;j\in S$$
(11)$$V_{ijt}\geq 0,\;\mathrm{for}\;i\in A,\;j\in S,\;t\in T.$$

The first term in the objective function ($\sum_{j}^{S}D_{j}C)$ calculates the total cost resulted from opening the selected sites. The second term, ($\sum_{i}^{A}\sum_{j}^{S}(\frac{1}{F}X_{ij}P_{i}(O_{i}TC_{ij}+\left(1-O_{i}\right)TB_{ij})\times 2\times VT))$ calculates the cost of time spent by people traveling to their closest F sites, assuming that households that own cars travel by car and households that do not own cars travel by public transit. 

Means of travel are accounted for by multiplying the percentage of people who own cars by car travel time (*TC_ij_*) and the remaining percentage by the public transit travel time (*TB_ij_*). The whole term is then multiplied by 2 to account for round trips and then by the travel time cost parameter VT to convert time into cost. The whole term is divided by F to average the possible F destination centers that could be used by each person. The third term ($\sum_{i}^{A}\sum_{j}^{S}(\frac{1}{F}X_{ij}P_{i}DD_{ij}O_{i}\times 2\times VD)$) finds the distance costs for car operation. It is multiplied by 2 to account for the round trip and by *VD*, which is the cost per mile. The fourth term ($\sum_{i}^{A}\sum_{j}^{S}(\frac{1}{F}X_{ij}P_{i}(1-O_{i})\times VB))$ finds the total cost spent by transit users by multiplying the number of trips by the roundtrip cost of transit ticket *VB*. 

Constraint (2) ensures an allocation strategy that treats all zones equally, whereby the ratio of the population of area *i* to the total population is multiplied by the available vaccine quantities *Q_t_* for the given period *t*. Constraint (3) sets the maximum allowable centers to MS. Constraint (4) sets the minimum possible centers to LS. Constraint (5) indicates that each person will receive their vaccine from one of the F nearest centers. Constraint (6) is an upper bound constraint to link decision variables *X_ij_* and *D_j_*. Constraint (7) sets the maximum budget for the costs of opening sites to K. In constraint (8), the amount of vaccines allocated is linked to the assignment between areas and sites so that vaccines are only allocated when there is an assignment, where *M* is an upper bound for *V_ijt_*. Constraints (9), (10), and (11) regulate the value of the decision variables.

### 2.2. Data

We applied the model in Section 2.1 to LA County, utilizing public data sets readily available in the United States, reflecting a vaccination campaign where the vaccine was available for the public free of charge and voluntarily. Our case study spanned a horizon of 12 biweekly periods, starting from the day the first vaccine was approved by the Food and Drug Administration (FDA) on 11 December 2020 [17] and extending until 30 May 2021, the actual number of vaccines administered in the county in each period. This period was chosen to focus on the initial phase of the pandemic when sites and vaccines were limited. Our analysis requires data on the demographic characteristics and locations of the sub-areas of metropolitan regions. Basic demographic characteristics of postal zip codes (population, proportional distribution by ethnicity, and age group) were sourced from the US Census Bureau [18]. Data from the 5-Year American Community Survey by the US Census Bureau [19] provided statistics on household car ownership per zip code, offering an indicator of how people might travel to vaccine centers. The Healthy Places Index (HPI) was sourced from the Public Health Alliance of Southern California [3]. It rates the zip codes of the county from 1 to 99, with higher scores indicating healthier regions. A COVID-19 vulnerability index was also developed based on CDC data [20], which provides rates of cases, hospitalizations, and fatalities by racial group. These rates were applied to individual zip codes based on their racial characteristics to produce a place-specific vulnerability index. Within the observed period, multiple COVID-19 vaccines were authorized to be used, including Pfizer-BioNTech, Moderna, and Johnson and Johnson. The frequency of vaccination is not directly modeled, but it is assumed to be captured by modeling the number of trips individuals make to receive their vaccinations. Vaccine administration data were sourced from the LA County Department of Public Health [21]. 

The value of time used in our analysis equals 50% of LA County’s average wage per minute, or USD 0.264 [22], as per the US Department of Transportation [23]. We used a driving cost per mile of USD 0.615, based on the American Automobile Association [24]. The public transit fare is USD 1.75 per trip. 

For visualization and spatial analysis, geographical data and shapefiles of LA County were sourced from the LA GeoHub [25]. Bing API was utilized to generate data on travel times and distances between LA County zip codes for both automobile and public transit. Utilizing population-weighted centroid from the Office of Policy Development and Research [26] coordinates, three 277 by 277 matrices were generated by Bing Maps for car travel time, transit travel time, and distance.

### 2.3. Scenarios and Weights

We explored three scenarios to demonstrate the effects of various prioritization methods. These scenarios help us understand the impact of different objectives on the outcomes. In Scenario One, the population alone served as the weighting factor in the objective function, treating all people equally. For this reason, the objective for Scenario One will be called unweighted. In Scenario Two, all areas were divided into five groups based on the HPI percentile, with each group representing 20% of the zones. The population of each area was multiplied by the weights in Table 1, which prioritizes zones with the lowest HPI. The middle zone (40th to 59th percentile) received a weight of one. 

In Scenario Three, the weighting factor *W_i_* was calculated based on the vulnerability of ethnicities to COVID-19 (Centers for Disease Control and Prevention, CDC [20]). Table 2 outlines the risks of infection, hospitalization, and death for COVID-19 by race and ethnicity. The “×” in these values shows the risk ratio compared to the White, non-Hispanic race. For instance, Black non-Hispanic individuals are 1.1 times more likely to be infected, two times more likely to be hospitalized and 1.6 times more likely to die from the virus than White individuals. These rates, combined with the racial compositions for the targeted areas of study, were used to calculate a COVID-19 racial vulnerability index. A weighted score for each of the three risks (cases, hospitalization, death) is computed by multiplying the rates from Table 2 by the proportion of each race for each area. 

We converted data derived from Table 2 into weights based on percentile groups, as with Scenario Two. First, the vulnerability cores (cases, hospitalizations, and deaths) were combined into a single score that accounts for disease severity (0.15, 0.3, and 0.55 multiplied by hospitalization, and death relative risk). Next, the computed risk for each zone was converted into five percentile groups, yielding *W_i_* from 1 to 5, as before. 

### 2.4. Modifications to MIP Formulation for Scenarios

The weights for Scenarios Two and Three change the objective function and the first constraint of the MIP. The rest of the formulation is unchanged. The modified objective function and first constraint are as follows:(12)$$\begin{array}{cc} Min\overset{S}{\underset{j}{\sum }}D_{j}C\;+ & \overset{A}{\underset{i}{\sum }}\overset{S}{\underset{j}{\sum }}(\frac{1}{F}X_{ij}W_{i}(O_{i}TC_{ij}+\left(1-O_{i}\right)TB_{ij})\times 2\times VT)\\ & +\overset{A}{\underset{i}{\sum }}\overset{S}{\underset{j}{\sum }}(\frac{1}{F}X_{ij}W_{i}DD_{ij}O_{i}\times 2\times VD)+\overset{A}{\underset{i}{\sum }}\overset{S}{\underset{j}{\sum }}(\frac{1}{F}X_{ij}W_{i}(1-O_{i})\times 2\times VB) \end{array}$$
(13)$$\overset{S}{\underset{j}{\sum }}V_{ijt}=\frac{W_{i}}{\sum Wi}\times Q_{t},\;\mathrm{for}\;i\in A,\;t\;\in T$$

## 3. Results: Case Study Application to LA County

As mentioned, LA County is the most populous county in the United States, with over 10 million residents. According to the US Census Bureau [18], the County consists of 49% Hispanic or Latino, 25.5% White, 7.6% Black, 14% Asian, and less than 1% American Indian and Alaska Native. Of the approximately 295 zip codes that make up LA County, 277 were considered for this study. This selection excluded zip codes with insufficient data and those on Catalina Island, given its unique accessibility constraints via ferries and airplanes. 

For our analysis, we assumed *C* and *K*, denoting that the site opening cost and total budget equaled USD 500,000 and USD 10 million, respectively. Actual costs should be based on real-life budget data. We solved the MIP using AMPL, which produced computation times of 70 s on an M2 MacBook Air.

### 3.1. Comparison of Costs and Travel Times

The three defined scenarios yielded varying results, highlighting the intricate balance between optimizing cost, serving highly populated areas, and prioritizing populations based on selected health and social vulnerabilities. Table 3 shows the total cost and its breakdown for all three scenarios, both unweighted (objective function for Scenario One) and measured according to each scenario’s objective (Objective Function Z), with F = 3, *MS* = 20, *LS* = 7, *K* = USD 10 million and *C* = USD 500,000. In all of our solutions, the optimal number of sites equaled MS, due to the relatively high transportation cost relative to construction cost.

The varying results across the three scenarios underscore trade-offs. Scenario One, when all people were weighted equally, yielded the lowest unweighted cost without considering other factors, such as health and social indices. On the other hand, Scenario Three produced the highest unweighted cost while lowering access costs for highly vulnerable people to the virus. Thus, while Scenario Three produced longer travel times and distances on average among all people, it reduced these costs for the most vulnerable people. Figure 1 shows the optimal locations for Scenario One. 

The first scenario concentrated vaccination sites in areas with dense populations in central Los Angeles. Many sites were selected in the downtown area and other dense urban zip codes. Nevertheless, some sites were located in the suburbs, which provided accessibility to residents living outside the central urban area, but at a greater distance, particularly in northern LA County. The second scenario also concentrated sites in the central zip codes, where the HPI index is relatively low compared to the outlying zip codes. However, more sites were located on the west side of the County than in the first scenario. A slight change in the suburban centers exists between the two scenarios. In Scenario Three, the locations are somewhat more scattered across the county.

The unweighted cost increase from Scenario One to Scenario Three shows the balance between minimizing travel costs on average versus preferentially serving those who are most vulnerable to disease. While prioritizing the average person might be cost-effective, it might not yield the best public health outcome. Table 4 provides average times and distances for each scenario. Because only Scenario One optimized this objective, it produced the lowest values. The other scenarios would produce optima relative to their scenario-specific weights.

The relationship between the average travel times by car and transit, as well as the average distance with the populations and population densities of the zip codes, are presented in scatterplots (Figure 2, Figure 3 and Figure 4):

The scatterplots show a consistent pattern: as the population and density of zip codes increase, the average travel times by car and transit, as well as the average distance, tend to decrease with lower variability. This pattern suggests that more densely populated areas benefit from shorter travel times and distances to vaccination sites.

### 3.2. Maximum Number of Vaccination Sites 

The model was tested with the number of allowed sites ranging between 7 and 40 for Scenario One. Figure 5 shows the objective function, including total costs and cost, by category. In each case, the number of selected sites equals the maximum allowed due to the relative importance of minimizing travel costs.

Starting at seven allowed sites, the total cost decreased as the maximum number of sites increased. The cost of opening sites increased linearly as more sites were added, which is expected since each additional site involves a fixed cost. The transit ticket cost is constant as the number of trips is unchanged and reflects the number of people who do not own cars, regardless of the number of sites. The travel time and distance costs decreased as more sites are opened, reflecting the reduced travel time and distances when more sites are spread across the county. The decrease in total cost and travel time and distance costs become less significant with each additional vaccination site. 

Figure 6 shows a trend of reduction in the average travel time by transit and car and the average distance traveled as the number of sites increases. A significant reduction is observed in the average travel time for transit users, which demonstrates the importance of opening more sites to increase accessibility for people who rely on public transit. A less steep decrease is seen for car users since they are generally more mobile and less affected by the number of available centers. Overall, the results demonstrated the advantage of allowing more vaccination sites, showing their benefits in terms of time and cost savings for the public.

Table 5 compares total costs for different maximum site values for all scenarios according to the unweighted objective. As the number of sites increased, the total costs declined, primarily due to the significant contribution of time and distance costs to the overall cost, with Scenario One producing the minimum cost in each case. 

### 3.3. Number of Sites People Choose from (F)

Our second analysis models the effects of increasing *F*, changing the number of sites each resident can choose from, with *MS* = 20. The model was run with *F* = 1 and *F* = 5 to test how F affects the selection of sites and the objective function. In Figure 7 and Figure 8, we showed the selected sites when *F* = 1 and *F* = 5, respectively, for Scenario One. 

As *F* increased, the model tended to cluster sites, as in Figure 8. If people need to search among multiple sites for vaccine availability, the advantage of increasing *MS* disappears, ultimately requiring people to travel further to receive a vaccine. In the extreme, when *F* = *MS*, all sites were close to each other. Table 6 provides the total weighted cost for *F* = 1, 3, and 5 and the three scenarios with *MS* = 20. The value of the objective function gets worse as *F* increases. Utilizing more choices might help individuals receive vaccines sooner but with the cost of longer trips. 

### 3.4. Comparison to the Actual LA County Vaccination Sites 

The model’s optimized solutions were compared to the actual mass vaccination sites in LA County. During the early phase of vaccination, nine locations (Figure 9) were established to serve county residents [1]. 

Although the sites were scattered across the region, some highly populated areas had no nearby sites, particularly in the northern and northeastern parts of LA County. For comparison, our model was applied with a maximum of nine sites, matching the actual number. Table 7 shows improvements of 16 to 18% in unweighted costs, travel times, and distances. Compared to the optimal solution, the sites in Figure 9 are spread less equally within the county and were not optimally situated to serve areas with the highest concentrations of people. 

### 3.5. Comparisons across Racial Groups for Nine Sites

We analyzed the average distance and travel time by racial group for all three scenarios as well as the actual selected sites. Table 8 and Table 9 show the results when *MS* = 9 and *F* = 1 and *F* = 3, respectively. In addition, Table 10, Table 11, Table 12 and Table 13 show the percentage of transit, car, and all travelers who spend more than 30 min traveling to get vaccinated (one way), representing those who experience more extreme travel times and distances. 

No single solution minimized travel time for all racial groups, which highlights the challenge of a one-size-fits-all approach when selecting vaccination sites. For instance, for *F* = 1, Scenario 2 produced the shortest travel time for Latinos, Black people, and American Indians; Scenario 3 produced the shortest travel time for Whites and Asian Americans; and Scenario 1 produced the shortest travel time overall. Comparing racial groups, Asian Americans experienced the lowest average travel times for the actual sites but not in our optimized solutions. Several actual sites were concentrated in San Gabriel Valley, where many Asians reside. In the optimized solutions, Latinos and Black people experienced the lowest travel times, due to their prevalence in central Los Angeles, where more sites were located when optimized. American Indians (AIs) and Whites tended to experience the longest travel times, even though they tended to live in zip codes where car ownership was higher. This can be attributed to a tendency to live in more remote and less densely populated locations. 

A significant disparity existed between transit and car users across all scenarios. Transit users spent more time traveling to sites, exceeding 30 min for the majority of riders in all scenarios and for all racial groups. For *F* = 3, nearly 100% of transit users, in all racial groups and all scenarios, exceed a 30 min travel time. This could be due to the vast area of LA County and the current public transportation infrastructure, which puts those dependent on public transit at a disadvantage. 

When F increased from 1 to 3, travel times increased, especially for transit users. For example, 59.3% of transit users spent more than 30 min to get to a site when *F* = 1 for Scenario One. This percentage increased to 84.1% when *F* = 3. The same trend can be observed across all scenarios. For car users, the percentage still increased, but less significantly. Increasing people’s options may provide greater access to vaccines but at the expense of longer trips. 

No single scenario universally benefited all racial groups. For Hispanics/Latinos, the travel time is shorter in Scenario Two when the HPI is utilized and worse in Scenario Three when the COVID-19 vulnerability index was used. Whites, on the other hand, experienced longer travel times for both car and transit, which could be explained by their residential patterns, where they often live in less populated suburban areas further away from the sites. 

Black people, in most cases, experienced shorter travel times than other racial groups. This could be due to their presence in urban areas, where populations are higher, and public transportation is more accessible and closer to vaccination sites. Asians in LA County generally resided within shorter travel times to get vaccinated than Whites. This may be due to higher average incomes, leading to more car ownership and a tendency to live in urban areas closer to vaccination sites. Lastly, American Indians tend to live in more rural areas where public transportation is limited and long car trips are needed, resulting in higher travel times to get vaccinated.

Our analysis demonstrates that all three proposed scenarios outperform the actual implemented plan for vaccination sites in terms of travel times, distances, and the overall percentage of people traveling more than 30 min, across all racial groups and the two modes of transportation. Significant and measurable improvements were achieved regardless of the choice to use population, HPI, or COVID-19 vulnerability as the weighting factor. These findings emphasize the pivotal role of data-driven strategies in enhancing the effectiveness and accessibility of public health resources.

## 4. Discussion

Using Los Angeles County as an example, this study illustrates that automobile ownership, population density, and site selection all affect access to mass vaccination sites. With a total of nine sites and optimized locations, the average travel time by car is approximately 20 min and by public transit is approximately 50 min. For the actual sites, average travel by car rose by about 5 min and by public transit by about 15 min. For both optimized and actual sites, travel times vary significantly by zip code, with the highest averages in zip codes with low population and low population density.

The more central areas of the county are the most densely populated areas and have lower automobile ownership. However, even in densely populated areas, the majority of households in LA County own automobiles. Central areas also tend to have higher proportions of the population who are Latino and Black. Our analysis accounted for all of these factors, producing travel time estimates by ethnic group and transportation model. All ethnic groups benefited from optimized site location, whether traveling by public transit or automobile. 

Our comprehensive scenario analysis revealed that in all three scenarios we implemented, the optimized sites selected led to measurable improvements in accessibility. When compared to the actual sites during the early phase of the pandemic, our optimized solution suggests a 24–29% improvement in travel time and distances, with Scenario One producing the most significant reductions in travel times by both car and transit across all racial groups. These findings were consistently observed even when the flexibility parameter F and the number of sites were varied, suggesting robustness in the model’s ability to enhance access under different conditions. 

The cost analysis revealed trade-offs associated with each objective. For example, Scenario One produced the minimum costs where people were all treated equally without considering health and social factors inherent to individual zip codes. On the other hand, Scenario Three yielded the highest cost but gave better access to locations where more people were highly vulnerable to the virus. 

The challenge in site selection is that no matter what solution is used, some people will live closer to sites than others. To produce the shortest average travel times, it is desirable to place sites closest to the largest concentrations of people, which, in LA County, brings sites closer to the urban center, where more Latinos and Blacks reside and fewer households own automobiles. On the other hand, adding more sites can reduce travel times to the closest site, efficiencies of mass vaccination may be lost, and people may need to search among more sites to obtain vaccines, ultimately adding to travel times (as when *F* increases). 

In our research, we assumed that the cost of opening a vaccination site is USD 500,000. This number is an approximation and does not consider the variability in actual real estate costs across different areas within LA County. The actual cost of establishing vaccination sites can vary significantly depending on the location, size, and other factors, such as rental, renovation, and operational costs. Accurate cost assessments in practice would involve financial analysis tailored for each site, incorporating detailed local real estate data, property size, and condition. We also assumed that every zip code contains a viable site for mass vaccination. Practical site selection may entail adjustments to nearby locations. Though the actual sites used in LA County were extremely large, such as a stadium and theme park, effective patient scheduling can greatly reduce the need for accommodating large queues, which accounted for much more of the utilized space than actual vaccine administration. Despite these limitations, our optimization model is easily adopted to site-specific considerations by adjusting parameters and utilizing specific site coordinates rather than zip code centroids. 

This study also did not consider the professional composition or age distribution of the population, which could lead to variability among zip codes in the requirements for vaccination. Occupation can significantly influence the individual’s risk of exposure to the virus as well as eligibility for vaccination. 

## 5. Conclusions

COVID-19 illustrates challenges in providing equitable and efficient access to vaccines at the scale of a metropolitan region. While establishing many sites for administering vaccines naturally reduces travel times to the nearest vaccination site, localized limits of vaccine supply may force individuals to travel to more distant locations. Thus, it may be advantageous to have fewer sites, each with assured supply, than many sites where availability is limited. In our model, as F increases, both travel time increases on average, and optimized sites converge into clusters. 

From the perspective of equity, from our analysis of LA County, optimized solutions tend to favor vaccination sites in densely populated areas toward the city center. Because these areas in LA County have higher concentrations of Latinos and Black people, they tended to have shorter travel times, even after factoring in access to automobiles at the household level. More explicitly, the HPI and COVID vulnerability did not change that outcome, though Latinos, Black people, and American Indians were slightly better off when HPI-based weights were used. On the other hand, people who do not have access to automobiles, regardless of race, are seriously disadvantaged, with much longer travel times. This effect is particularly strong when F is greater than one. While a compelling argument for having many vaccination sites is access for those dependent on public transit, that advantage disappears if people must search among many sites to find a location with available vaccines. 

Future research could potentially account for localized and current metrics of disease prevalence (e.g., daily or weekly rates of cases and deaths by zip code). The practicality of such an approach is limited by the data aggregation of statistics, which have been reported at the community level rather than zip codes, along with signification fluctuations in case and death rates from day to day. Nevertheless, sites might be more precisely located to serve areas experiencing disease at the highest rates if public health agencies could track data within consistent and small geographic units. Future models could also improve the accuracy of the costs of establishing sites by utilizing more precise real-time data of real estate, which would account for the variability of site costs across different areas. Moreover, the professional decomposition of the population was not considered in our current model, which could help improve access to frontline workers as they have higher risks of exposure. Lastly, public transit stops and usage data per zip codes were not considered in the study, which could be a strong tool to significantly improve the travel times for transit users and enhance their accessibility.

## Figures and Tables

**Figure 1 ijerph-21-00491-f001:**
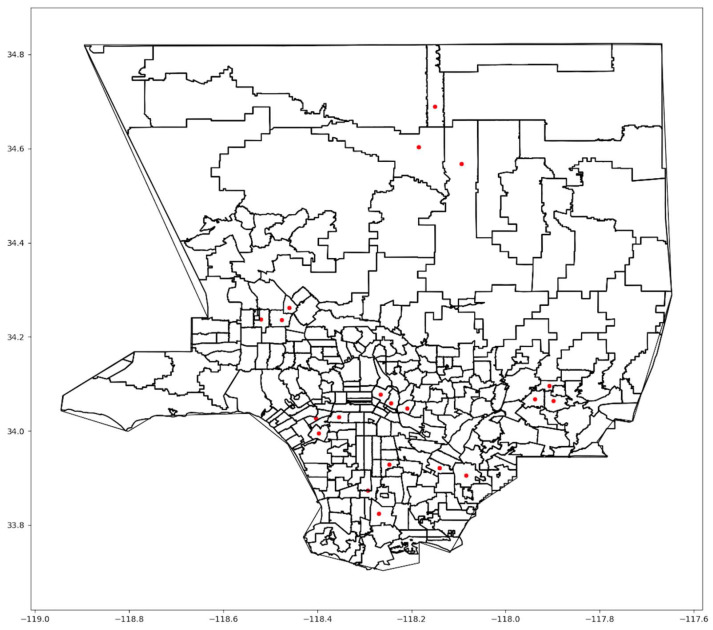
Scenario One optimal locations.

**Figure 2 ijerph-21-00491-f002:**
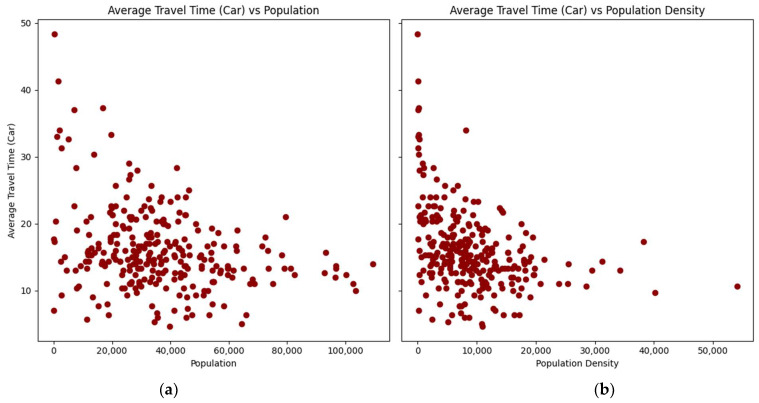
Scatterplots of average travel time by car by zip code: (**a**) versus population; (**b**) versus population density.

**Figure 3 ijerph-21-00491-f003:**
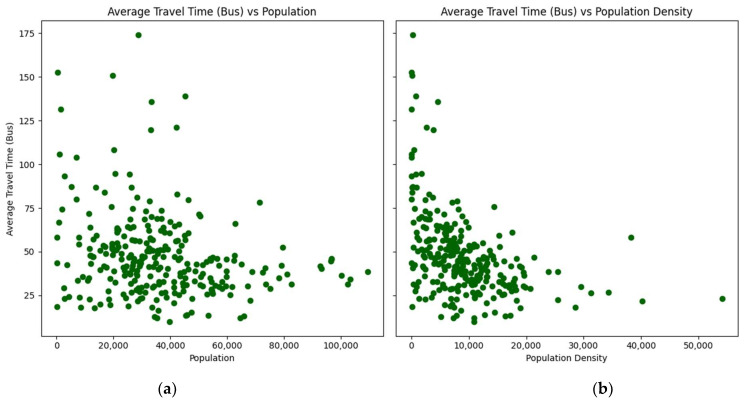
Scatterplots of average travel time by transit by zip code: (**a**) versus population; (**b**) versus population density.

**Figure 4 ijerph-21-00491-f004:**
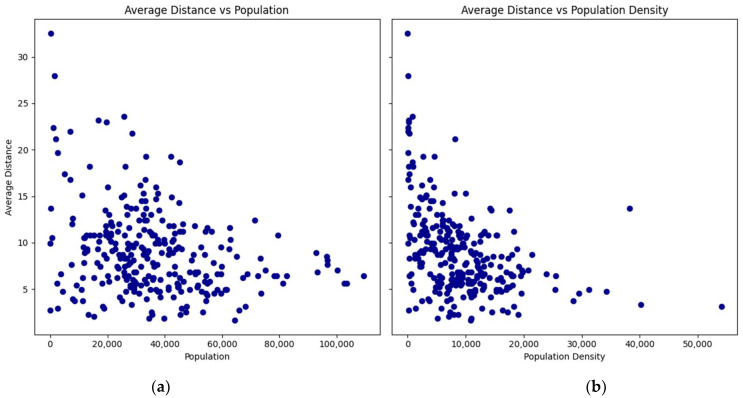
Scatterplots of average distance by zip code: (**a**) versus population; (**b**) versus population density.

**Figure 5 ijerph-21-00491-f005:**
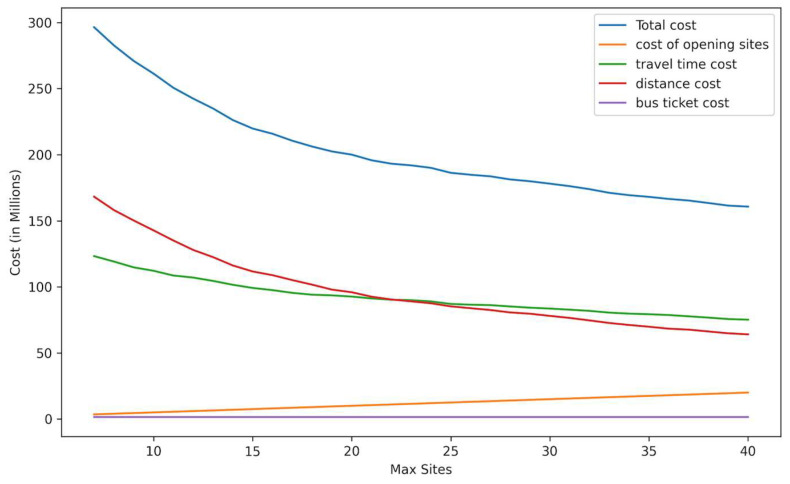
Cost by type (million USD) vs. maximum number of sites.

**Figure 6 ijerph-21-00491-f006:**
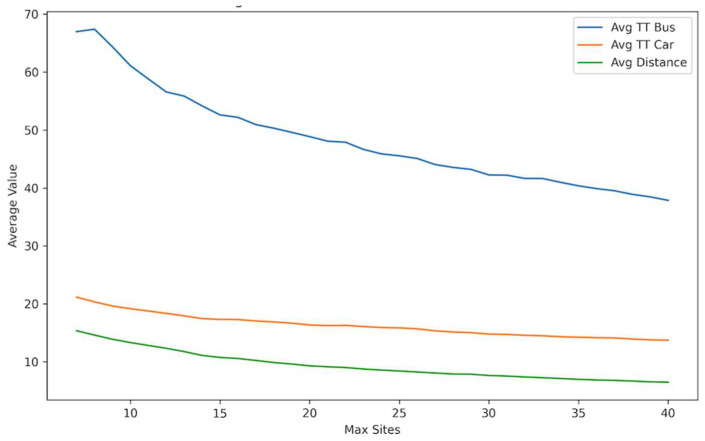
Average travel times and distance vs. maximum number of sites.

**Figure 7 ijerph-21-00491-f007:**
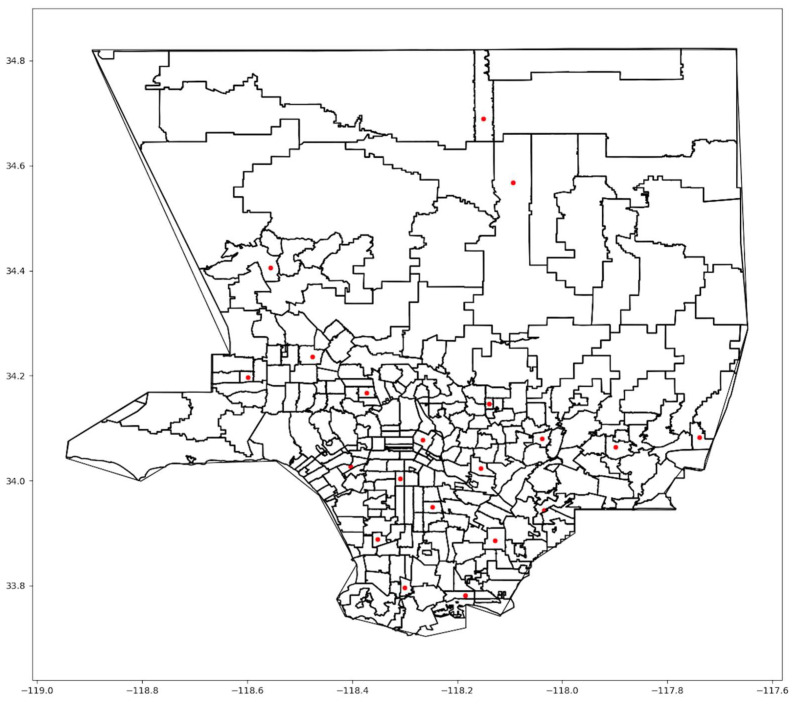
Map of optimal sites when *F* = 1.

**Figure 8 ijerph-21-00491-f008:**
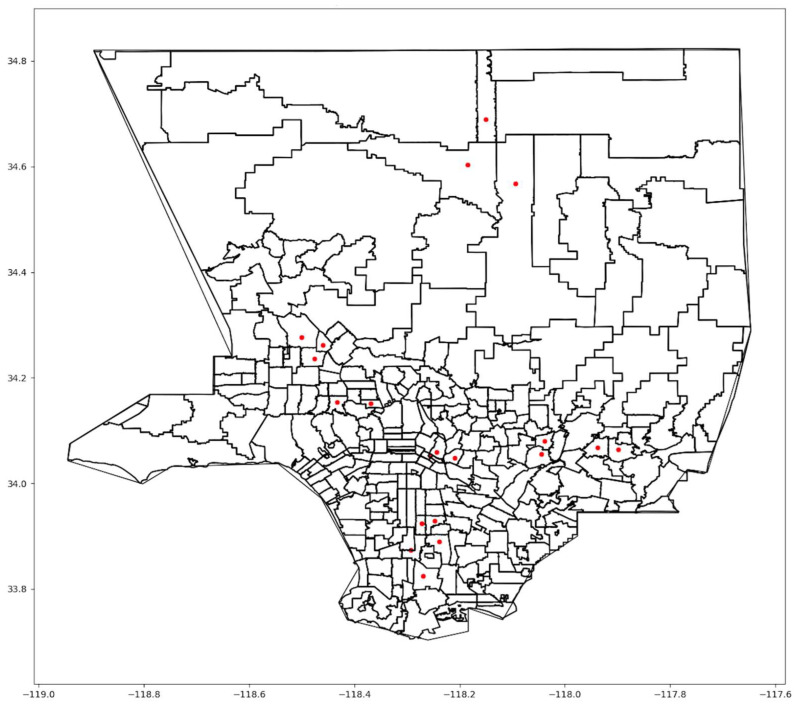
Map of optimal sites When *F* = 5.

**Figure 9 ijerph-21-00491-f009:**
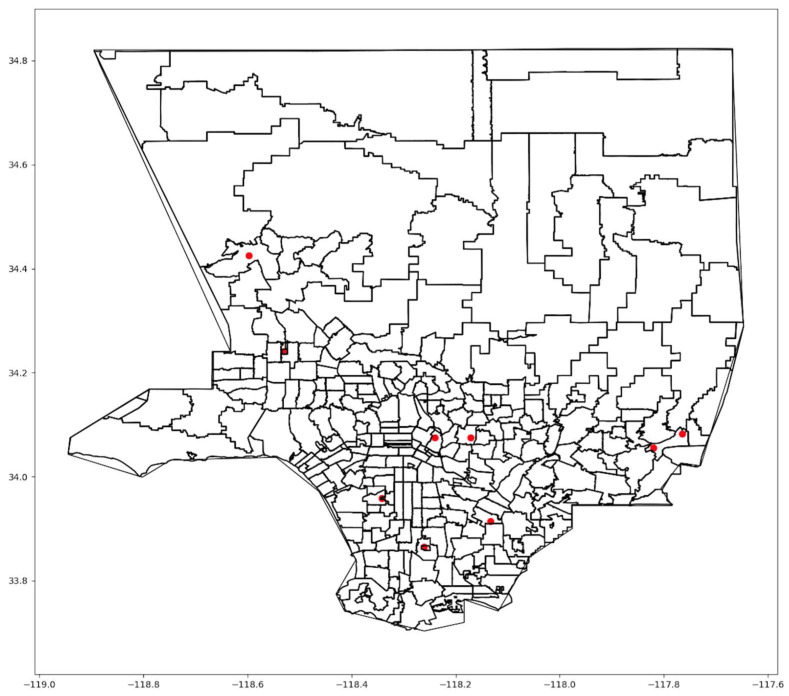
Map of actual mass vaccination sites in LA County.

**Table 1 ijerph-21-00491-t001:** *Wi* calculations using HPI.

Percentile Range	Category	Weighted Population (*W_i_*)
0.19–0	1	population × 1.5
0.39–0.2	2	population × 1.25
0.59–0.4	3	population × 1
0.79–0.6	4	population × 0.75
1–0.8	5	population × 0.5

**Table 2 ijerph-21-00491-t002:** Racial vulnerability to COVID-19 by CDC.

Race/Ethnicity	American Indian or Alaska Native, Non-Hispanic	Asian, Non-Hispanic	Black or African American, Non-Hispanic	Hispanic or Latino
Cases	1.6×	0.8×	1.1×	1.5×
Hospitalization	2.4×	0.7×	2.0×	1.8×
Death	2.0×	0.7×	1.6×	1.7×

**Table 3 ijerph-21-00491-t003:** Cost breakdown by scenario in million USD.

Scenario	Construction Cost	Travel Time Cost	Travel Distance Cost	Transit Tickets Cost	Total Cost (Unweighted)	Objective Function Z (Weighted)
1	10	89.6	93.7	1.49	194.8	194.8
2		90.3	94.6		196.4	204
3		92.6	95.9		200	182.3

**Table 4 ijerph-21-00491-t004:** Average travel times and distances for the scenarios in minutes and miles.

Scenario	Avg Travel Time by Car	Avg Travel Time by Transit	Avg Distance
1	14.9	44.1	8.2
2	15	44.7	8.3
3	15.3	46.1	8.5

**Table 5 ijerph-21-00491-t005:** Total unweighted cost for different max sites for all scenarios in millions USD.

Scenario	Maximum Sites Number				
	7	15	20	25	35
1	293.5	215	194.8	181.4	164.2
2	293.5	215.9	196.4	184.4	166.9
3	296.4	219.8	200	186.3	168.1

**Table 6 ijerph-21-00491-t006:** Total weighted costs for different F values for all scenarios in million USD.

Scenario	Flexibility Parameter (*F*)		
	1	3	5
1	122.7	194.8	245.3
2	125.9	196.4	248.2
3	127.5	200	247

**Table 7 ijerph-21-00491-t007:** Comparison of scenarios to actual sites (unweighted).

Scenario	Avg Travel Time by Car	Avg Travel Time by Transit	Avg Distance	Travel Time Cost	% Improvements in Avg Values	Distance Cost	Total Cost (Including Time, Distance, and Opening)	% Savings in Costs
1	64.6	19.6	13.9	113 M	18	149.5 M	268.5 M	17
2	64.4	19.6	13.9	113.7 M	18	149.4 M	269 M	17
3	67.3	19.6	13.9	114.7 M	17	150 M	270.1 M	16
Actual Sites	79.9	25.3	15.7	147.7 M	-	170 M	323.7 M	-

**Table 8 ijerph-21-00491-t008:** Average times and distance for racial groups when *MS* = 9 and *F* = 1.

	Average Travel	Average for All Races	Latino	White	Black	American Indian	Asian
Scenario One	Car (Time Min.)	13.6	12.9	15.4	12.8	14.0	13.5
	Transit (Time Min.)	34.1	33.1	37.8	31.6	34.0	33.6
	Overall (Time Min.)	15.4	14.8	17.0	14.7	15.7	15.1
	Car Distance (Miles)	7.0	6.5	8.3	6.5	7.4	6.3
Scenario Two	Car (Time Min.)	13.8	12.6	16.0	12.5	13.9	14.4
	Transit (Time Min.)	32.1	30.5	37.3	28.4	31.7	33.1
	Overall (Time Min.)	15.3	14.2	17.6	14.1	15.5	15.9
	Car Distance (Miles)	7.1	6.3	8.7	6.2	7.4	7.6
Scenario Three	Car (Time Min.)	14.3	14.6	14.3	14.3	14.8	13.4
	Transit (Time Min.)	37.4	39.3	35.9	35.6	37.4	33.5
	Overall (Time Min.)	16.3	16.9	15.9	16.5	16.8	15.0
	Car Distance (Miles)	7.2	7.5	7.2	7.1	7.5	6.0
Actual LA Sites	Car (Time Min.)	19.3	18.6	21.6	19.0	22.8	17.5
	Transit (Time Min.)	47.0	45.3	54.7	43.1	57.0	44.1
	Overall (Time Min.)	21.7	21.1	24.0	21.5	25.7	19.6
	Car Distance (Miles)	9.4	9.1	10.8	9.4	12.8	8.0

**Table 9 ijerph-21-00491-t009:** Average times and distance for racial groups when *MS* = 9 and *F* = 3.

	Average Travel	Average for All Races	Latino	White	Black	American Indian	Asian
Scenario One	Car (Time Min.)	18.6	17.8	20.4	18.5	22.5	18.2
	Transit (Time Min.)	51.1	49.0	58.0	51.9	62.1	48.1
	Overall (Time Min.)	21.4	20.6	23.1	21.9	25.8	20.5
	Car Distance (Miles)	13.2	12.5	14.8	13.3	17.5	12.8
Scenario Two	Car (Time Min.)	18.7	17.7	20.5	18.4	21.2	18.6
	Transit (Time Min.)	51.7	49.2	58.6	51.7	58.6	50.6
	Overall (Time Min.)	21.5	20.6	23.2	21.9	24.4	21.1
	Car Distance (Miles)	13.2	12.4	14.8	13.2	16.1	13.2
Scenario Three	Car (Time Min.)	18.7	18.2	20.1	18.9	22.6	17.6
	Transit (Time Min.)	54.0	52.2	60.2	54.9	66	50.0
	Overall (Time Min.)	21.7	21.3	23.0	22.7	26.3	20.2
	Car Distance (Miles)	13.3	12.9	14.6	13.9	17.8	12.1
Actual LA Sites	Car (Time Min.)	24.3	23.4	26.8	24.3	27.9	22.9
	Transit (Time Min.)	66.8	64.1	76.7	61.7	77.6	64.9
	Overall (Time Min.)	27.9	27.2	30.4	28.1	32.1	26.2
	Car Distance (Miles)	15.1	14.4	17.1	15.1	19	13.9

**Table 10 ijerph-21-00491-t010:** Percentage of entire population spending more than 30 min to vaccination site (*MS* = 9).

		% of Transit Users Spending above 30 min One Way	% of Car Users Spending above 30 min One Way	Overall % of Both
*F* = 1	SC 1	59.34%	0.51%	5.45%
	SC 2	57.22%	0.59%	5.35%
	SC 3	62.66%	0.65%	5.86%
	Actual	76.10%	5.53%	11.39%
*F* = 3	SC 1	84.08%	5.07%	11.65%
	SC 2	88.31%	5.00%	11.94%
	SC 3	90.75%	5.46%	12.56%
	Actual	99.22%	12.22%	19.37%

**Table 11 ijerph-21-00491-t011:** Percentage of transit users spending more than 30 min by racial group (MS = 9).

		% of Transit Riders Spending above 30 Min One Way				
		Latino	White	Black	AI	Asian
*F* = 1	SC 1	59.7%	63.8%	57.6%	54.0%	51.7%
	SC 2	54.6%	66.8%	49.4%	52.1%	57.8%
	SC 3	67.6%	57.5%	65.7%	58.6%	48.6%
	Actual	75.2%	86.8%	54.0%	78.2%	77.9%
*F* = 3	SC 1	82.2%	90.1%	78.7%	82.3%	86.0%
	SC 2	86.4%	93.7%	85.1%	89.1%	89.5%
	SC 3	90.4%	91.9%	89.7%	91.0%	91.0%
	Actual	98.9%	99.9%	98.3%	99.9%	99.9%

**Table 12 ijerph-21-00491-t012:** Percentage of car users spending more than 30 min by racial group (*MS* = 9).

		% of Transit Riders Spending above 30 Min One Way				
		Latino	White	Black	AI	Asian
*F* = 1	SC 1	0.24%	1.32%	0.17%	0.89%	0.18%
	SC 2	0.26%	1.56%	0.19%	0.89%	0.22%
	SC 3	0.55%	1.24%	0.23%	1.57%	0.16%
	Actual	4.81%	8.25%	8.29%	15.01%	1.70%
*F* = 3	SC 1	4.94%	6.25%	8.40%	14.29%	1.65%
	SC 2	4.89%	6.05%	8.39%	14.22%	1.64%
	SC 3	5.29%	6.87%	8.61%	14.45%	1.81%
	Actual	8.95%	21.15%	12.02%	21.22%	7.80%

**Table 13 ijerph-21-00491-t013:** Percentage of entire population spending more than 30 min by racial group (*MS* = 9).

		% of Transit Riders Spending above 30 Min One Way				
		Latino	White	Black	AI	Asian
*F* = 1	SC 1	5.2%	5.6%	5.4%	5.2%	4.0%
	SC 2	4.9%	6.0%	4.5%	5.0%	4.6%
	SC 3	6.4%	5.2%	6.7%	6.2%	3.8%
	Actual	11.0%	13.2%	12.7%	20.3%	7.4%
*F* = 3	SC 1	11.9%	12.3%	15.6%	19.9%	8.0%
	SC 2	12.3%	12.4%	16.3%	20.6%	8.5%
	SC 3	12.8%	12.6%	16.8%	20.8%	8.5%
	Actual	16.9%	25.3%	20.6%	26.7%	14.3%

## Data Availability

All data analyzed and examined in this study are presented within this article.

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
