# Peer review of "Optimizing the Selection of Mass Vaccination Sites: Access and Equity Consideration"

_ijerph, 2024, doi:10.3390/ijerph21040491_

Round 1

Reviewer 1 Report

Comments and Suggestions for Authors

Dear Authors:

Great job putting together this interesting piece. The COVID-19 pandemic era affected everyone in different ways. The advent of the COVID-19 vaccine to help manage the health implications of the infection was a ray of hope. However, access became an issue affecting everyone that wanted the vaccine differently. And I agree, access can be tied to issues around equity.

I have a few comments and recommendations for your consideration:

Lines 32-33: Viruses are not treatable. COVID-19 is not a treatable disease, instead it’s a preventable disease. And the COVID-19 vaccine and other oral medications were not designed to treat, rather were designed to help mitigate the impact of the virus. They were developed to reduce the health implications of contracting the virus, such as hospitalizations and death from COVID-19. Consider rephrasing this sentence to align with the right info, providing evidence from the literature.

Lines 40 – 41: consider substantiating this claim with tenable references.

Lines 42 – 44: a repetition in the research objectives subsection. Consider removing this.

Lines 52 – 53: consider including a reference to substantiate this claim.

Lines 45 – 63: 1.1. Research objectives: there is a lot of repetition in this subsection.  This section appears to mirror subsection 1.3. Research contribution. Consider collapsing these two subsections into one section and paragraph.

Lines 65 – 66: Consider including relevant references to substantiate this claim.

 Materials & Methods:

What specific time period is being studied? This will help us understand the relevance of the findings to the current reality.

Parameter Dj was not defined under the list of parameters. Consider including a definition for Dj.

Results:

Figure 2,3 & 4: Consider including a sentence that states the implication of the patterns from the scatterplots.

Did not find a discussion section. I understand the discussion contents appeared in the result section. Consider delineating the result and discussion. And consider including tenable references to support your findings (in tandem with what is or is not obtainable elsewhere or with other types of infectious diseases).

What are some of the limitations? You may use the last paragraph of the discussion section (after revision) to highlight potential limitations of the conduct of this study.

Author Response

Dear reviewer, 

We appreciate the input and feedback.  We have explained how we have responded to individual comments in the attached file. 

Thank you,

Reviewer 2 Report

Comments and Suggestions for Authors

I really admire following finding and it certainly help the policy makers,

1.      Site Distribution and Vaccine Supply: While having numerous vaccination sites can reduce travel times for individuals, localized shortages of vaccine supply may necessitate travel to more distant locations. Therefore, it might be more advantageous to have fewer sites with guaranteed vaccine supply than many sites with limited availability.

2.      Impact of Optimization: As the number of vaccination sites (denoted as F) increases, both average travel time and optimized site locations tend to converge into clusters.

3.      Equity Considerations: Optimized solutions tend to favor vaccination sites in densely populated areas near the city center, which often have higher concentrations of Latinos and Black people. However, access to vaccines for those without access to automobiles, regardless of race, remains a significant challenge, especially when the number of vaccination sites is high.

4.      Cost Considerations: The study assumes a cost of $500,000 for opening a vaccination site, but actual costs may vary based on factors such as location, size, rental, renovation, and operational costs. Additionally, practical site selection may require adjustments to nearby locations.

Overall, the findings underscore the complex interplay between site distribution, vaccine supply, equity considerations, and cost implications in ensuring widespread access to COVID-19 vaccination.

Questions:

1.      What specific challenges does the excerpt highlight regarding equitable access to COVID-19 vaccines in metropolitan regions like Los Angeles County?

2.      How do the findings suggest a trade-off between the number of vaccination sites and the efficiency of vaccine distribution?

3.      Can you elaborate on how optimized solutions tend to favor vaccination sites in densely populated areas, and what implications this has for equity in vaccine access?

4.      How do factors such as race and access to automobiles influence individuals' travel times to vaccination sites?

5.      What are some potential adjustments or considerations that could be made in practical site selection to address the challenges identified in the study?

6.      Could you provide more information on the assumptions made regarding the cost of opening vaccination sites and how this might impact decision-making in vaccine distribution strategies?

7.      In what ways do the findings of this study contribute to our understanding of the logistical and equity challenges associated with COVID-19 vaccination efforts in urban areas?

Author Response

(The authors gave the same response as above.)

Reviewer 3 Report

Comments and Suggestions for Authors

Thank you for the opportunity to review the manuscript. The research topic is relevant because coronavirus infection still poses a large-scale problem for healthcare systems around the world. I have several comments on this study: 1) the approach chosen by the authors to assess the availability of vaccination in the population is somewhat mechanistic. It includes an assessment of the distance of vaccination centers, population density and ethnic composition. In this regard, I have a question whether the authors studied the socio-demographic characteristics of the population (age ranking, since elderly and senile people have less access to medical services, but at the same time have less contact with patients with COVID-19), professional composition (for example, medical staff or teachers are in greater need of vaccination due to high risks of infection); 2) it is unclear which vaccines were used to vaccinate the population included in the study; 3) whether the comorbidity of the population was taken into account during vaccination; 4) what was the frequency of vaccination; whether vaccination was paid or free, mandatory or voluntary. It seems to me that the authors should have involved a specialist in the field of healthcare organization to work on the manuscript to cover the topic more comprehensively.

Author Response

(The authors gave the same response as above.)

Round 2

Reviewer 1 Report

Comments and Suggestions for Authors

Dear Authors,

This is a more improved version of the former.

One more recommendation (adapted from my previous comments):

Section 1.2. Literature review. "Optimization models have been increasingly utilized in healthcare to improve health outcomes and increase efficiency and effectiveness." This is a statement of fact. Are there studies that have showed optimization model improved health outcomes? Then, consider citing published work that supports this claim. 

Section 3. Results. Consider moving the first 2 sentences of paragraph 2 to the method section.

Discussion: Are there studies (from different settings or that took into account entirely different outbreaks) with findings that are similar to or different from findings of the current study? Consider including them, while also providing appropriate references.

Comments on the Quality of English Language

Consider conducting a thorough grammar and spelling check. For example: 

Section 2. Materials and Methods. This section should be reported as past tense.

Line 1 - consider replacing "formulate" with "formulated," so that it can now read as "In this section, we formulated a ..."

Author Response

Dear reviewers, 

We appreciate the input and feedback. Please find our response to the comments in the attached file. 

Thank you,
